# Trajectories and correlates of opioid prescription receipt among patients experiencing interpersonal violence

Jessica R. Williams[1]*, Ishrat Z. Alam[2,3], Shabbar I. Ranapurwala[2,3]

**1** School of Nursing, University of North Carolina at Chapel Hill, Chapel Hill, North Carolina, United States of America, **2** Department of Epidemiology, Gillings School of Public Health, University of North Carolina at Chapel Hill, Chapel Hill, North Carolina, United States of America, **3** Injury Prevention Research Center, University of North Carolina at Chapel Hill, Chapel Hill, North Carolina, United States of America

* jrober65@email.unc.edu

**Data Availability Statement:** Data cannot be shared publicly due to concerns of patient privacy. Data are available from the North Carolina Translational and Clinical Science Institute

## Abstract

Interpersonal violence increases vulnerability to the deleterious effects of opioid use. Increased opioid prescription receipt is a major contributor to the opioid crisis; however, our understanding of prescription patterns and risk factors among those with a history of interpersonal violence remains elusive. This study sought to identify 5-year longitudinal patterns of opioid prescription receipt among patients experiencing interpersonal violence within a large healthcare system and sociodemographic and clinical characteristics associated with prescription patterns. This secondary analysis examined electronic health record data from January 2004–August 2019 for a cohort of patients (N = 1,587) referred for interpersonal violence services. Latent class growth analysis was used to estimate trajectories of opioid prescription receipt over a 5-year period. Standardized differences were calculated to assess variation in sociodemographic and clinical characteristics between classes. Our cohort had a high prevalence of prescription opioid receipt (73.3%) and underlying co-morbidities, including chronic pain (54.6%), substance use disorders (39.0%), and mental health diagnoses (76.9%). Six prescription opioid receipt classes emerged, characterized by probability of any prescription opioid receipt at the start and end of the study period (high, medium, low, never) and change in probability over time (increasing, decreasing, stable). Classes with the highest probability of prescription opioids also had the highest proportions of males, chronic pain diagnoses, substance use disorders, and mental health diagnoses. Black, non-Hispanic and Hispanic patients were more likely to be in low or no prescription opioid receipt classes. These findings highlight the importance of monitoring for synergistic co-morbidities when providing pain management and offering treatment that is trauma-informed, destigmatizing, and integrated into routine care.

## Introduction

Prescription and non-prescription opioid use disorder and overdose is a national public health crisis: an estimated 115 people die every day due to opioid overdose [1]. Despite efforts to reduce opioid prescriptions, the number of individuals who receive prescriptions each year

(nctracs@unc.edu) and University of North Carolina at Chapel Hill Institutional Review Board (irbreliance@unc.edu) for researchers who meet the criteria for access to confidential data.

**Funding:** Research reported in this publication was fully supported by an Exploratory Research Project grant (JRW) through an Injury Control Research Center award (R49-CE003092) from the Centers for Disease Control and Prevention (CDC), National Center for Injury Prevention and Control. Additionally, SIR and IZA were funded through R01 CE003009 funded by the CDC. The findings and conclusions in this publication are those of the authors and do not necessarily represent the views of the CDC. The funders had no role in the study design, data collection and analysis, decision to publish, or preparation of the manuscript.

**Competing interests:** The authors have declared that no competing interests exist.

remains high [2]. Opioid prescription receipt greatly increases a person's risk for adverse outcomes, particularly when prescribed long-term; approximately 21–29% of individuals prescribed long-term opioid therapy for chronic pain use them non-medically [3] and 8–12% develop an opioid use disorder (OUD) [4, 5]. A better understanding of factors associated with long-term opioid prescription receipt, particularly among high-risk populations, can help facilitate efforts to prevent OUD.

Consistent with other substance use disorder research, evidence shows that rather than a single linear trajectory, opioid use follows multiple trajectories characterized by significant heterogeneity in behaviors [6–9]. Prior research has examined opioid use trajectories among diverse groups of patients including veterans, patients receiving care for HIV, patients enrolled in a Medicaid lock-in program, and individuals receiving treatment for OUD [9–12]. Opioid use trajectories have been shown to vary by amount of use (e.g., low, medium, high) and use patterns over time (e.g., increasing use, decreasing use). Trajectory patterns are associated with numerous factors including demographics, comorbid conditions, substance use history, and social/family conditions. These relationships, however, vary based on the population studied.

One population particularly vulnerable to deleterious effects of opioid use is individuals with a history of interpersonal violence. Experiences of interpersonal violence (e.g., intimate partner violence, sexual assault), have been linked to an increased likelihood of prescription opioid use, [13–15] non-medical prescription opioid use, [13, 14] OUD, [16–18] and heroin use [19]. Opioid prescription receipt has also been found to increase odds of OUD in this population. A recent study found that among individuals with a history of interpersonal violence, people who had received an opioid prescription in the past year had a four times greater odds of reporting opioid misuse behaviors compared to those without an opioid prescription [20].

Evidence also suggests that interpersonal violence may influence opioid use trajectories: two studies showed that physical, sexual, and emotional abuse history influences recovery trajectories for individuals seeking treatment for substance use disorders (including OUD), with higher abuse levels predicting more severe psychiatric and social problems and slower recovery times [9, 21]. Although providing some evidence about the role of interpersonal violence in opioid use trajectories, these studies were conducted after treatment for a diagnosed substance use disorder and did not differentiate by type of substance. Thus, our understanding of prescription opioid use patterns among individuals with a history of interpersonal violence remains elusive. Identifying distinct patterns of prescription opioid use among survivors of interpersonal violence and associated characteristics will help us understand who is at risk for escalating opioid use and OUD and provide potential targets for future interventions.

The purpose of this study was to examine longitudinal trajectories of prescription opioid use among patients experiencing interpersonal violence. Specifically, we aimed to:

*Aim 1.* Classify 5-year longitudinal patterns of opioid prescription receipt among patients experiencing interpersonal violence.

*Aim 2.* Identify sociodemographic and clinical characteristics associated with opioid prescription receipt patterns among survivors of interpersonal violence.

## Methods

### Design

We conducted a secondary analysis of electronic health record (EHR) data available from January 2004–August 2019 for a cohort of patients referred for interpersonal violence services

between December 2009–June 2019 at the University of North Carolina (UNC) Health System (N = 1,587). We used latent class growth analysis (LCGA) to estimate trajectories of opioid prescription receipt over a 5-year period. We then examined associations between trajectory patterns and sociodemographic and clinical characteristics. This study was approved by the University of North Carolina Institutional Review Board (#20–0988).

## Study population

The study population consisted of all adult patients ($\geq$ 18 years of age at baseline) referred to the UNC Hospitals Beacon Program for interpersonal violence services between December 2009–June 2019 with at least one year of data in the EHR during the study period. The Beacon Program provides comprehensive, coordinated care to UNC Health System patients, families, and employees experiencing a interpersonal violence, including intimate partner violence, human trafficking, sexual assault, and elder/vulnerable adult abuse. Individuals were excluded if they had evidence of an OUD at baseline (diagnosis or buprenorphine, methadone, or naltrexone prescription), had a cancer diagnosis, or were receiving end-of-life care during the study period as opioid prescribing practices differ for these conditions.

## Variables

All data analyzed in this study were electronically abstracted from the UNC Health System EHR databases. This included data related to opioid prescription receipt and sociodemographic and clinical characteristics. Given that experiences of interpersonal violence are often chronic rather than an isolated event, we did not limit data extraction based on date of referral to the Beacon Program, but rather extracted relevant variables across the study period (January 2004–August 2019).

**Opioid prescription receipt.** We examined prescription drug records for opioid medications prescribed during the study period. Then, we recorded all opioid prescriptions during the 60 months following baseline. We focused on the following opioids commonly prescribed for pain: butorphanol, codeine, dihydrocodeine, fentanyl, hydrocodone, hydromorphone, levorphanol, meperidine, morphine, opium, oxycodone, oxymorphone, pentazocine, propoxyphene, tapentadol, and tramadol. Dosage and duration information was not readily available from the EHR data, especially prior to April 2014; thus, we could not calculate morphine milligram equivalents for prescriptions. Therefore, outcome variable was defined as a binary indicator of any prescription opioid receipt for each month of the study period.

**Socio-demographic and clinical characteristics.** Demographic characteristics were assessed at the beginning of the study period and included age, race, ethnicity, and sex. We examined insurance status across the study period, categorizing patients as insured, not insured, or varying status. We used International Classification of Diseases 9th and 10th edition (ICD-9 and ICD-10) codes to identify medical diagnoses during the study period. Diagnoses included chronic pain conditions, substance use disorders, and mental health conditions. We calculated the Charlson comorbidity index, a method of categorizing comorbidities based on ICD codes, for each patient [22, 23]. Lastly, we collected information on social history, namely, smoking and illicit drug use history. Social history is routinely screened for by providers within the UNC Health System and documented within each patient's EHR. Specific information on diagnostic codes used to define characteristics is presented in S1 Table.

## Statistical analyses

LCGA was used to examine opioid prescription receipt over a 5-year time period and identify groups with similar trajectories (Aim 1). Baseline was defined as date of first opioid

prescription during the study period (January 2004–August 2019). Those who did not have an opioid prescription during the study period were placed in a "Never" class. We fit models with 2 to 7 groups to determine the optimal number of trajectories. Model selection was based on a combination of model fit, adequacy, and interpretability. We assessed fit using Akaike's information criterion (AIC), [24] Bayesian information criterion (BIC), [25] bootstrap likelihood ratio test (BLRT), and entropy. The LCGA was fitted using MPlus, version 8.4.

To estimate the association between sociodemographic and clinical characteristics with trajectory groups, we estimated the prevalence of patient characteristics within each latent class. The analysis was weighted using patient's posterior probabilities for belonging to a given class. We estimated and graphed standardized differences to assess the degree of similarity and dissimilarity between classes within a given covariate ("Never" class used as reference group). The analysis was conducted in Stata SE, version 6.1.

## Results

### Baseline population characteristics

Between December 2009 and June 2019, 4,296 individuals were referred to the Beacon Program for interpersonal violence services. We excluded 2,709 people resulting in a final sample size of 1,587. The primary reason for exclusion was < 1 year of data in the EHR. Cohort characteristics are presented in Table 1. Our cohort was largely female (87.5%), White, non-Hispanic (55.2%) or Black, non-Hispanic (29.3%), insured (77.3%), and had a mean age of 41.4 years (SD = 16.3). Over half (58.3%) had a current smoking history and almost half (44.8%) had a lifetime history of illicit drug use. Our cohort exhibited a high prevalence of chronic pain (54.6%), substance use disorders (39.0%), and mental health diagnoses (76.9%) during the study period. The most common chronic pain diagnoses were chronic low back pain (33.5%) and chronic pain, not elsewhere classified (30.0%). Alcohol (18.0%) and cannabis (16.1%) related disorders were the most prevalent substance use diagnoses; however, rates were similar across all drug related diagnoses. Notably, 13.0% received an OUD diagnosis after baseline (participants were excluded if OUD diagnosis was present at baseline). The most common mental health related diagnoses included depression (65.5%) and anxiety related disorders (53.9%). The mean Charlson comorbidity index was 1.4 (SD = 2.1), which corresponds to an estimated 10-year survival of 94% [23, 26].

### Longitudinal patterns of opioid prescription receipt among patients experiencing interpersonal violence

Table 2 shows the model fitting results for the LGCA. Four-hundred-twenty-three patients (26.7%) did not receive an opioid prescription during the study period and, thus, were not included in the LGCA. Fit indices favored solutions with an increasing number of classes. The BLRT behaved similarly, yielding significant test results comparing a *K*-class model relative to a *K*—1-class model with an increasing number of classes before becoming unstable at a 7-class solution. We selected a 6-class model for further analyses based on fit indices, interpretability, class size, and stability.

Fig 1 shows trajectories of opioid prescription receipt based on estimated probability over time. We characterized the trajectory classes based on beginning probability, ending probability, and degree of change as follows:

• Never: no opioid prescription receipt (26.7%)

• Class 1: Start medium, end medium, moderate decrease (4.8%)

**Table 1. Characteristics of study participants, and by latent class of prescription opioid trajectories: Dec 2009- Jun 2019.**

| | Total cohort who received Beacon program assistance (n = 1,587)[a] | Never prescribed opioids (n = 423.0, 26.7%)[a] | Opioid prescription receipt trajectories developed using LCGA analysis | | | | | |
|---|---|---|---|---|---|---|---|---|
| | | | Class 1 | Class 2 | Class 3 | Class 4 | Class 5 | Class 6 |
| | | | Start medium, end medium, moderate decrease (n = 75.4, 4.8%)[b] | Start high, end high, slight increase (n = 32.4, 2.0%)[b] | Start medium, end low, sharp decrease (n = 206.2, 13.0%)[b] | Start low, end high, sharp increase (n = 51.1, 3.2%)[b] | Start low, end medium, moderate increase (n = 216.6, 13.6%)[b] | Start low, end low, stable (n = 582.3, 36.7%)[b] |
| Age (mean, SD) | 41.4 (16.3) | 39.4 (17.4) | 43.4 (14.9) | 47.3 (15.5) | 42.5 (14.9) | 41.9 (15.5) | 40.7 (15.6) | 42.0 (16.4) |
| Sex | | | | | | | | |
| Male | 198 (12.5) | 51 (12.1) | 15.2 (20.1) | 4.4 (13.7) | 20.3 (9.8) | 7.1 (14.0) | 27.8 (12.9) | 72.2 (12.4) |
| Female | 1,389 (87.5) | 372 (87.9) | 60.2 (79.9) | 27.9 (86.3) | 185.9 (90.2) | 44.0 (86.1) | 188.7 (87.2) | 510.2 (87.6) |
| Race/ Ethnicity | | | | | | | | |
| White, non-Hispanic | 847 (55.2) | 210 (52.6) | 48.1 (64.0) | 23.4 (72.2) | 119.4 (59.2) | 29.2 (60.0) | 116.7 (55.2) | 300.2 (53.0) |
| Black, non-Hispanic | 449 (29.3) | 101 (25.3) | 20.6 (27.4) | 7.9 (24.5) | 62.3 (30.9) | 13.5 (27.8) | 69.8 (33.0) | 173.9 (30.7) |
| Hispanic | 163 (10.6) | 61 (15.3) | 5.9 (7.9) | 1.0 (3.1) | 12.4 (6.2) | 3.4 (7.0) | 15.0 (7.1) | 64.3 (11.4) |
| Other, non-Hispanic | 76 (5.0) | 27 (6.8) | 0.5 (0.7) | 0.6 (0.2) | 7.6 (3.8) | 2.6 (5.3) | 10.1 (4.8) | 28.2 (5.0) |
| Insurance | | | | | | | | |
| Insured | 1,150 (77.3) | 301 (78.6) | 58.9 (80.7) | 25.7 (80.3) | 152.0 (77.8) | 37.0 (76.3) | 156.7 (76.4) | 418.6 (76.1) |
| Not insured | 166 (11.2) | 39 (10.2) | 5.6 (7.7) | 1.1 (3.5) | 21.5 (11.0) | 4.1 (8.5) | 22.7 (11.1) | 72.0 (13.1) |
| Varying | 171 (11.5) | 43 (11.2) | 8.5 (11.6) | 5.2 (16.2) | 22.0 (11.3) | 7.3 (15.1) | 25.7 (12.5) | 59.3 (10.8) |
| Smoking History | | | | | | | | |
| Never | 347 (21.9) | 129 (30.6) | 8.4 (11.2) | 6.3 (19.4) | 35.6 (17.3) | 6.5 (12.8) | 37.1 (17.1) | 124.1 (21.3) |
| Now | 923 (58.3) | 212 (50.4) | 49.6 (66.0) | 19.8 (61.2) | 134.0 (65.2) | 29.2 (57.2) | 136.6 (63.1) | 341.8 (58.7) |
| Past | 314 (19.8) | 80 (19.0) | 17.1 (22.8) | 6.3 (19.4) | 36.0 (17.5) | 15.4 (30.1) | 42.8 (19.8) | 116.4 (20.0) |
| Illicit Drug—Ever | 707 (44.8) | 186 (44.3) | 36.0 (48.6) | 15.1 (46.7) | 83.4 (40.6) | 23.6 (46.3) | 109.0 (50.4) | 253.9 (43.7) |
| *Chronic pain* | | | | | | | | |
| Chronic Pain, not elsewhere classified | 476 (30.0) | 37 (8.8) | 59.2 (78.5) | 28.5 (88.1) | 95.5 (46.3) | 32.9 (64.4) | 87.9 (40.6) | 135.0 (23.2) |
| Chronic pain syndrome | 138 (8.7) | 5 (1.2) | 26.6 (35.3) | 13.3 (41.1) | 24.8 (12.0) | 19.7 (38.6) | 24.1 (11.1) | 24.4 (4.2) |
| Fibromyalgia | 202 (12.7) | 15 (3.6) | 21.2 (28.1) | 10.8 (33.5) | 35.4 (17.2) | 10.2 (20.0) | 41.4 (19.1) | 68.1 (11.7) |
| Irritable Bowel Syndrome | 58 (3.7) | 4 (1.0) | 7.2 (9.5) | 2.0 (6.2) | 13.5 (6.5) | 3.4 (6.6) | 9.6 (4.4) | 18.4 (3.2) |
| Interstitial cystitis/ Bladder pain syndrome | 15 (1.0) | 0 (0.0) | 1.1 (1.4) | 1.0 (3.1) | 2.4 (1.2) | 1.2 (2.0) | 5.4 (2.5) | 4.1 (0.7) |
| Migraine | 249 (15.7) | 33 (7.8) | 24.3 (32.3) | 8.1 (25.1) | 40.8 (19.8) | 10.6 (20.8) | 47.5 (21.9) | 84.6 (14.5) |
| Chronic low back pain | 532 (33.5) | 63 (14.9) | 45.3 (60.1) | 22.0 (68.0) | 90.5 (43.9) | 27.6 (53.9) | 103.5 (47.8) | 180.1 (30.9) |
| Chronic Fatigue Syndrome | 28 (1.8) | 2 (0.5) | 2.1 (2.8) | 0.0 (0.0) | 3.2 (1.6) | 3.8 (7.5) | 4.9 (2.3) | 12.0 (2.1) |
| Endometriosis | 45 (2.8) | 1 (0.2) | 3.1 (4.1) | 2.0 (6.2) | 9.0 (4.4) | 5.6 (11.0) | 12.6 (5.8) | 11.7 (2.0) |
| *Substance use disorders* | | | | | | | | |
| Alcohol related disorders | 286 (18.0) | 67 (15.8) | 20.6 (27.3) | 4.3 (13.3) | 41.7 (20.2) | 10.2 (19.9) | 43.3 (20.0) | 99.0 (17.0) |
| Opioid related disorders | 206 (13.0) | 29 (6.9) | 17.6 (23.3) | 9.1 (28.2) | 39.8 (19.3) | 16.3 (31.9) | 34.9 (16.1) | 59.3 (10.2) |
| Cannabis related disorders | 256 (16.1) | 56 (13.2) | 16.5 (21.8) | 3.1 (9.5) | 38.0 (18.4) | 10.9 (21.4) | 45.5 (21.0) | 86.1 (14.8) |

*(Continued)*

**Table 1.** (Continued)

| | Total cohort who received Beacon program assistance (n = 1,587)[a] | Never prescribed opioids (n = 423.0, 26.7%)[a] | Opioid prescription receipt trajectories developed using LCGA analysis | | | | | |
|---|---|---|---|---|---|---|---|---|
| | | | Class 1 | Class 2 | Class 3 | Class 4 | Class 5 | Class 6 |
| | | | Start medium, end medium, moderate decrease (n = 75.4, 4.8%)[b] | Start high, end high, slight increase (n = 32.4, 2.0%)[b] | Start medium, end low, sharp decrease (n = 206.2, 13.0%)[b] | Start low, end high, sharp increase (n = 51.1, 3.2%)[b] | Start low, end medium, moderate increase (n = 216.6, 13.6%)[b] | Start low, end low, stable (n = 582.3, 36.7%)[b] |
| Sedative, hypnotic, or anxiolytic related disorders | 113 (7.1) | 17 (4.0) | 11.5 (15.2) | 2.0 (6.2) | 22.7 (11.0) | 9.2 (18.1) | 16.5 (7.6) | 34.1 (5.9) |
| Cocaine related disorders | 229 (14.4) | 35 (8.3) | 16.3 (21.7) | 7.4 (22.9) | 39.4 (19.1) | 8.1 (15.9) | 42.1 (19.4) | 80.6 (13.8) |
| Other psychoactive substance related disorders | 209 (13.2) | 47 (11.1) | 15.0 (19.9) | 7.2 (22.2) | 35.2 (17.1) | 11.2 (21.9) | 28.5 (13.2) | 64.9 (11.2) |
| Unspecified/other drug dependence | 178 (11.2) | 18 (4.3) | 17.2 (22.8) | 5.6 (17.4) | 32.6 (15.8) | 9.3 (18.3) | 35.8 (16.6) | 59.4 (10.2) |
| *Mental health* | | | | | | | | |
| Depression | 1,040 (65.5) | 257 (60.8) | 60.1 (79.7) | 28.0 (86.6) | 142.5 (69.1) | 40.0 (78.2) | 153.9 (71.1) | 358.5 (61.6) |
| Schizophrenia, schizotypal, delusional, and other non-mood psychotic disorders | 212 (13.4) | 70 (16.6) | 12.8 (17.0) | 3.0 (9.4) | 29.0 (14.0) | 8.8 (17.2) | 24.0 (11.1) | 64.4 (11.1) |
| Anxiety, dissociative, stress related, somatoform and other non-psychotic mental disorders | 856 (53.9) | 195 (46.1) | 53.8 (71.3) | 22.9 (70.6) | 130.1 (63.1) | 39.1 (76.6) | 129.2 (59.7) | 285.9 (49.1) |
| Disorders of adult personality and behavior | 206 (13.0) | 50 (11.8) | 17.6 (23.3) | 6.0 (18.6) | 30.4 (14.8) | 9.9 (19.3) | 31.6 (14.6) | 60.5 (10.4) |
| Other mood disorders | 417 (26.3) | 101 (23.9) | 32.6 (43.3) | 16.6 (51.2) | 63.1 (30.6) | 15.6 (30.6) | 60.5 (27.9) | 127.6 (21.9) |
| Charlson comorbidity index (mean, SD) | 1.4 (2.1) | 0.8 (1.6) | 2.4 (2.5) | 2.8 (2.7) | 1.7 (2.1) | 2.4 (2.7) | 1.7 (2.2) | 1.3 (2.0) |

[a]Counts, percentages, and means are unweighted.

[b]Counts, percentages, and means are weighted by patient's posterior probabilities for belonging to a given class.

SD = standard deviation.

**Table 2. Model fit statistics (n = 1,164).**

| Number of Classes | AIC | BIC | Adjusted BIC | BLRT | | Entropy |
|---|---|---|---|---|---|---|
| | | | | Test Statistic | *p* value | |
| 1 Class | 31860.484 | 31870.603 | 31864.251 | - - | - - | - - |
| 2 Class | 26787.922 | 26813.220 | 26797.339 | 5078.562 | 0.0000 | 0.931 |
| 3 Class | 25646.498 | 25686.975 | 25661.565 | 1147.424 | 0.0000 | 0.889 |
| 4 Class | 25176.558 | 25232.213 | 25197.274 | 475.941 | 0.0000 | 0.830 |
| 5 Class | 24885.560 | 24956.395 | 24911.926 | 296.997 | 0.0000 | 0.758 |
| 6 Class | 24689.233 | 24775.247 | 24721.249 | 202.327 | 0.0000 | 0.738 |
| 7 Class | 24585.536 | 24686.728 | 24623.201 | 109.698 | 0.0000 | 0.757 |

*Note*. AIC = Akaike information criteria; BIC = Bayesian information criteria; BLRT = Bootstrap Likelihood Ratio Test.

- Class 2: Start high, end high, slight increase (2.0%)

- Class 3: Start medium, end low, sharp decrease (13.0%)

- Class 4: Start low, end high, sharp increase (3.2%)

- Class 5: Start low, end medium, moderate increase (13.6%)

- Class 6: Start low, end low, stable (36.7%)

Most participants clustered around patterns indicative of no or minimal opioid prescription probability (63.4%, Never and Class 6). The remaining classes, which characterize higher probabilities of opioid prescription receipt, demonstrated considerable variability over time. The two classes that began with a medium probability (Class 1 and Class 3), both showed decreases over time. The only class that began with a high probability (Class 2), remained high over time and slightly increased. Two classes that started low (Class 4 and 5), demonstrated increases over time.

### Sociodemographic and clinical characteristics associated with opioid prescription receipt trajectories

A number of sociodemographic and clinical characteristics were associated with opioid prescription receipt trajectories (Table 1, Fig 2). A summary of the key characteristics for each trajectory group is provided in S2 Table. Overall, prescription opioid classes (Classes 1–6) demonstrated higher prevalence for smoking, chronic pain, substance use disorders, mental

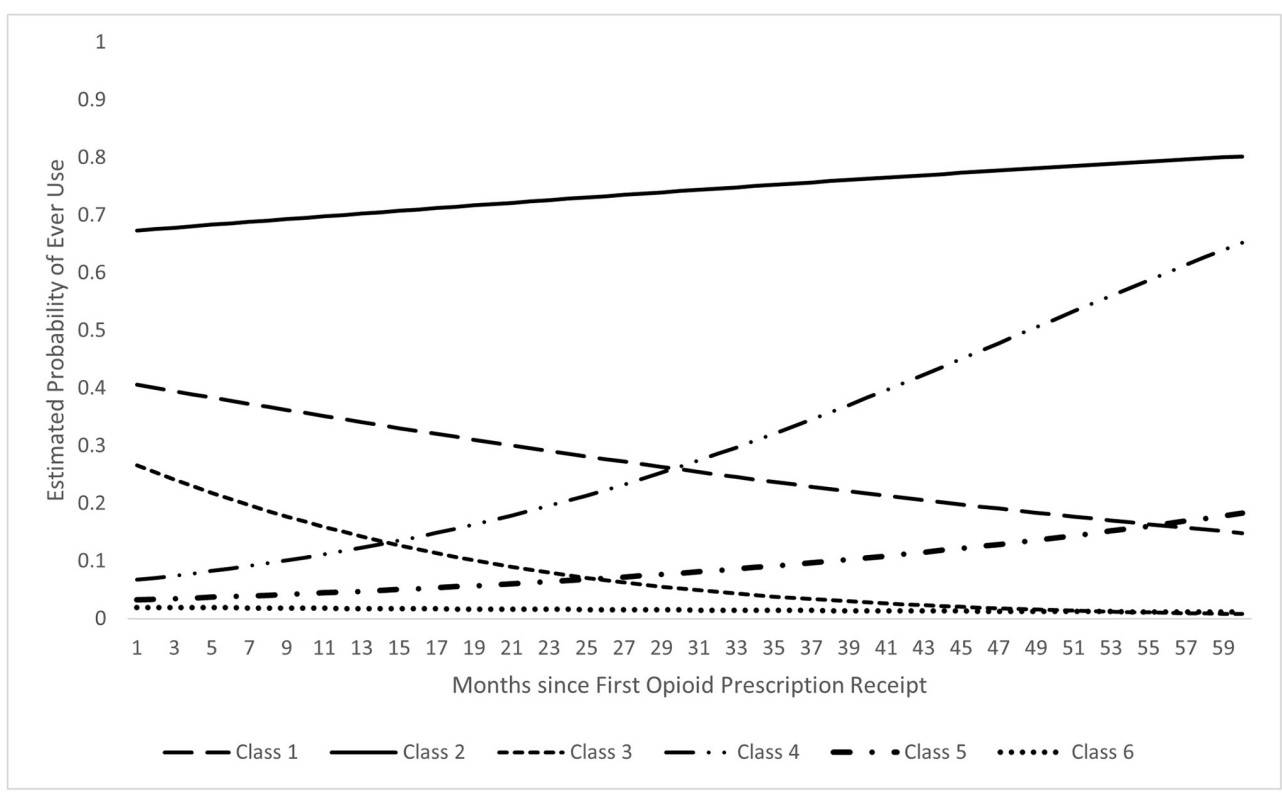

**Fig 1. Estimated probability of opioid prescription receipt by latent class trajectories.**

health disorders, and Charlson comorbidity index compared with the Never class. Never class were also the youngest among all classes. Class 6 (start low, end low) was most like the Never class, demonstrating the smallest standardized mean differences. The largest proportion of White, non-Hispanic patients was seen in the classes with the highest probability of prescription opioids (Class 1, 2, and 4). Males were also more likely to be represented in these classes whereas the highest proportion of females were in Class 3 (start medium, end low, sharp decrease). Black, non-Hispanic patients were more likely to be in low prescription opioid classes (Class 3, 5, and 6) and Hispanic and Other, non-Hispanic patients in the Never class. The largest proportion of insured patients were seen in higher prescription opioid classes (Class 1 and 2). Those with varying insurance status were also more likely to be represented in high prescription opioid classes (Class 2 and 4). Classes 1, 2, and 4, which demonstrated the highest probabilities of opioid use, had the highest proportions of chronic pain diagnoses, substance use disorders, including OUD, and mental health diagnoses.

**Comparison of Class 4 and Class 5.**  Individuals who clustered around Class 4 and Class 5 both started with a low probability of opioid prescription; however, Class 4 demonstrated a sharp increase in the probability of opioid prescription over time whereas Class 5 demonstrated a more moderate increase. We compared these two classes to identify factors that may contribute to the increased opioid prescription trajectory for Class 4. Both classes had similar demographics except Class 5 had a higher proportion of Black, non-Hispanics. Class 4 had a higher prevalence of chronic pain diagnoses (except for interstitial cystitis/bladder pain syndrome and migraine) and mental health disorders. Class 4 also had a higher prevalence of substance use disorders.

**Comparison of Class 1 and Class 3.**  Class 1 and Class 3 both started with a medium probability of prescription opioid use; Class 3 demonstrated a sharp decline over time and Class 1 demonstrated a more moderate decrease. We compared these two classes to identify factors that may contribute to the sharper decline in opioid prescription trajectory for Class 3. Both classes had similar demographics except Class 3 had a higher proportion of females. Class 3 had a lower prevalence of most pain diagnoses (except endometriosis and interstitial cystitis/bladder pain syndrome), substance use disorders, and mental health diagnoses compared to Class 1.

## Discussion

This study is the first to examine longitudinal prescription opioid trajectories among a cohort of patients with a history of interpersonal violence. Almost three-quarters (73.3%) received an opioid prescription during the study period (January 2004–August 2019). We identified six distinct patterns of opioid prescriptions over time, varying by starting and ending probability (high, medium, low, never) and degree of change over time (increase, decrease, stable). These findings indicate significant heterogeneity in opioid prescriptions over time among those impacted by interpersonal violence. This heterogeneity echoes findings from prior research that have modelled opioid use trajectories among other high-risk populations and provides further evidence of the presence of clinically significant subgroups of people who use prescription opioids [9–12].

Sociodemographic and clinical characteristics varied by prescription opioid trajectory class. Males were more likely to be represented in classes with the highest probability of being prescribed opioids (Class 1, 2, & 4) and the highest proportion of females was seen in Class 3 (start medium, end low, sharp decrease). These findings are consistent with prior research showing that women are more likely to be prescribed and use lower doses of opioids compared to men; [27] however, contrast with other studies indicating that women, in general, are more

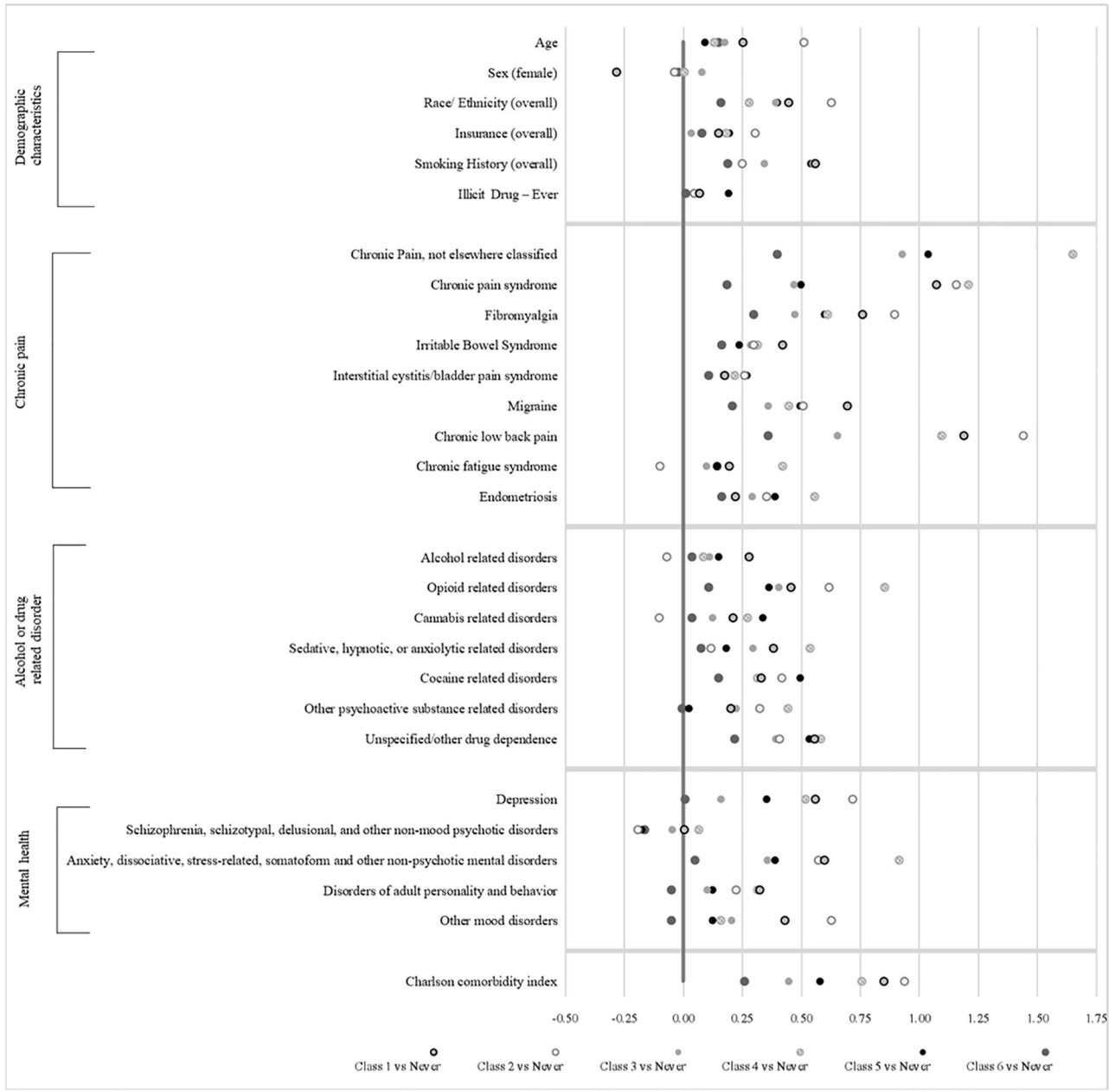

**Fig 2. Standardized mean differences in patient characteristics by latent class.**

likely to be prescribed opioids [28–31]. The higher proportion of males in the higher prescription opioid use classes may be problematic given that prior research indicates that men are at higher risk of misusing opioids [31–33]. The potential role of interpersonal violence on these differences is unclear. Only one study has examined sex differences in prescription opioid use among individuals with a history of interpersonal violence and did not find significant differences between males and females [34]. Additional work is needed to understand better sex differences in the relationship between interpersonal violence and prescription opioid use.

Race was also associated with class membership in our study. Specifically, Black, non-Hispanic patients were more likely to be in low prescription opioid classes (Class 3, 5, and 6) and Hispanic and Other, non-Hispanic patients in the Never class. This is consistent with the abundance of research highlighting racial/ethnic disparities in opioid prescribing practices through the U.S. [35–38]. Specifically, individuals of racial/ethnic minority statues are less likely than Whites to be prescribed opioids for pain and more likely to have pain symptoms underestimated by providers [35–38]. These disparities in pain management can be attributed to mistrust and biases by providers; [39, 40] negative experiences that often influence a patient's engagement in healthcare services, quality of patient-provider communication, and health outcomes [41–44]. These consequences may be exacerbated in patients with a history of interpersonal violence, which has also been shown to lead to discomfort, perceived discrimination, stigma, and mistrust in the healthcare environment [45–47]. Additional research is needed to understand better the intersection of interpersonal violence and racial/ethnic bias on the pain experience and to develop evidence-based approaches to mitigate their effects on pain management.

There was high prevalence of underlying co-morbidities within our cohort including chronic pain (54.6%), substance use disorders (39.0%), and mental health diagnoses (76.9%). The classes with the highest probability of prescription opioids (Classes 1, 2, and 4) also had the highest proportions of chronic pain diagnoses, substance use disorders, and mental health diagnoses. These co-morbidities also appeared to contribute to escalating risk of opioid use over time when comparing Class 4 to Class 5 and Class 1 to Class 3. Our findings also echo prior literature indicating that long-term prescription opioid use increases risk for OUD, both in the general population and among those with a history of IPV [3, 20]. In our cohort, 13.0% received an OUD diagnosis after baseline and this proportion was markedly higher among classes with higher probability of prescription opioid receipt over time (e.g., Class 4 = 31.9%; Class 2 = 28.2%; Class 1 = 23.3%). Prior research estimates that, in the general population, about 8–12% of people who are prescribed opioids will develop an OUD [3–5]. While we are unable to determine if the increased rates of OUD in our study are a result of new OUD cases or increased monitoring/identification, our findings provide some preliminary evidence that IPV may increase risk for the development of OUD among those prescribed opioids. Additional research is needed to better understand this potential risk.

The high rates of co-morbidities found in this study, especially among classes with higher probability of prescription opioid receipt over time, is consistent with the abundance of research demonstrating that chronic pain, substance use, and mental health conditions are common consequences of interpersonal violence [48–51]. More recently, researchers have begun to establish that these consequences are not independent of each other, but rather interact synergistically (often referred to as syndemic health conditions [52]), resulting in worsening health [20, 50, 53, 54]. Given this, it may be particularly important to monitor for synergistic co-morbidities when providing pain management and offer treatment that is trauma-informed, destigmatizing, and integrated into routine care.

A major limitation, however, in providing effective pain management for patients impacted by interpersonal violence is a lack of trauma-informed pain management strategies. Trauma-informed care provides a model to assist health providers understand, recognize, and respond to the effects of trauma on patients' health [55]. Interventions based on a trauma-informed approach have shown promise for improving patient outcomes in a variety of settings; [56–62] however, limited evidence exists on the impact of trauma-informed care on pain-related outcomes. Additional research is needed to examine how current evidence-based pain management practices can be tailored to better address the specific challenges (e.g., comorbid conditions) survivors of interpersonal violence face during their path to recovery.

## Limitations

There are several limitations, which should be acknowledged. 1) Our sample was limited to UNC Health Care System patients identified as experiencing interpersonal violence and referred for services. This misses those who may be experiencing interpersonal violence but were not identified or referred, thus reducing our generalizability. In addition, our study was conducted in one health care system in North Carolina, which also impacts generalizability. 2) Given the secondary nature of our data, we were not be able to determine the frequency, duration, or severity of interpersonal violence, which has been shown in prior work to impact opioid use [34]. 3) We were only able to examine longitudinal trends in opioid prescription receipt, and unable to examine trends in prescription dosage among people affected by interpersonal violence. This is important because prior studies show that higher doses and longer durations of opioid receipt are associated with OUD and overdose. Our study findings suggest that people with exposure to interpersonal violence are at high risk for both opioid use and OUD and may, therefore, be an important factor to consider when prescribing opioids. However, without information on morphine milligrams equivalent or days' supply, it is difficult to identify those who may specifically benefit from an opioid prescribing-related intervention. 4) Lastly, we examine the probabilities of receiving an opioid prescription for pain management, which may overestimate the people who actually fill and consume those prescription opioids. However, given the high prevalence of pain, substance use disorders, mental health disorders, and Charlson morbidity index among those receiving opioid prescriptions in this study, it is feasible to expect that vast majority of these prescriptions may have been filled and consumed.

## Conclusions

Results from this exploratory study suggest that people with exposure to interpersonal violence are at high risk for prescription opioid use (>70%) and OUD (13%). The different trajectories may help target interventions to people with interpersonal violence exposure who are at the highest risk for OUD (trajectories 1, 2, and 4). The high rates of co-morbid conditions among trajectories with higher probability of prescription opioid receipt over time highlight the importance of monitoring for co-morbidities among patients receiving care for chronic pain and the need for improved trauma-informed pain management strategies.

## Supporting information

**S1 Table. International classification of diseases 9th and 10th edition (ICD-9, ICD-10) codes used to define model covariates.**
(DOCX)

**S2 Table. Summary of key characteristics for each opioid prescription trajectory class.**
(DOCX)

## Acknowledgments

The authors acknowledge the support of Bill Ross and the NC Translational and Clinical Science (NC TraCS) Institute in assisting with data extraction from the UNC Health Care System electronic health records. NC TraCS is supported by the National Center for Advancing Translational Science (NCATS), National Institutes of Health, through Grant Award Number UL1TR002489.

## Author Contributions

**Conceptualization:** Jessica R. Williams, Shabbar I. Ranapurwala.

**Data curation:** Ishrat Z. Alam.

**Formal analysis:** Ishrat Z. Alam.

**Funding acquisition:** Jessica R. Williams.

**Methodology:** Jessica R. Williams, Shabbar I. Ranapurwala.

**Project administration:** Jessica R. Williams.

**Supervision:** Jessica R. Williams.

**Writing – original draft:** Jessica R. Williams, Ishrat Z. Alam.

**Writing – review & editing:** Jessica R. Williams, Ishrat Z. Alam, Shabbar I. Ranapurwala.

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
