## [Decision Letter · Decision Letter 0]

6 Jun 2022

PONE-D-22-04197

Trajectories and correlates of opioid prescription receipt among patients experiencing interpersonal violence

PLOS ONE

Dear Dr. Williams,

Thank you for submitting your manuscript to PLOS ONE. After careful consideration, we feel that it has merit but does not fully meet PLOS ONE’s publication criteria as it currently stands. Therefore, we invite you to submit a revised version of the manuscript that addresses the points raised during the review process, including requests to clarify aspects of the methodology and elaborate on the clinical significance of results.  

We look forward to receiving your revised manuscript. Thank you for submitting this important work to Plos One.

Kind regards,

Judith I Tsui

Academic Editor

PLOS ONE

**Journal requirements:**

“The authors acknowledge the support of Bill Ross and the NC Translational and Clinical Science (NC TraCS) Institute in assisting with data extraction from the UNC Health Care System electronic health records. NC TraCS is supported by the National Center for Advancing Translational Science (NCATS), National Institutes of Health, through Grant Award Number UL1TR002489.”

“Research reported in this publication was fully supported by an Exploratory Research Project grant (JW) through an Injury Control Research Center award (R49-CE003092) from the Centers for Disease Control and Prevention (CDC), National Center for Injury Prevention and Control. The findings and conclusions in this publication are those of the authors and do not necessarily represent the views of the CDC. The funders had no role in the study design, data collection and analysis, decision to publish, or preparation of the manuscript.”

5. Please note that in order to use the direct billing option the corresponding author must be affiliated with the chosen institute. Please either amend your manuscript to change the affiliation or corresponding author, or email us at plosone@plos.org with a request to remove this option.

Reviewers' comments:

Reviewer's Responses to Questions

**Comments to the Author**

1. Is the manuscript technically sound, and do the data support the conclusions?

Reviewer #1: Partly

Reviewer #2: Partly

2. Has the statistical analysis been performed appropriately and rigorously? 

Reviewer #1: Yes

Reviewer #2: I Don't Know

3. Have the authors made all data underlying the findings in their manuscript fully available?

Reviewer #1: No

Reviewer #2: Yes

4. Is the manuscript presented in an intelligible fashion and written in standard English?

Reviewer #1: Yes

Reviewer #2: Yes

5. Review Comments to the Author

Reviewer #1: This study examined trajectories of prescription opioid receipt and associated baseline characteristics among patients who were referred to interpersonal violence services in an academic healthcare system. This study is potentially of interest to researchers and clinicians, however the manuscript has some weaknesses. The analysis is limited by not being able to distinguish high-risk opioid prescribing from any prescription opioid receipt, and this limitation should be more robustly addressed. Additionally, some aspects of the methods need clarification. Finally, it is not clear from the discussion what implications these findings have for clinical practice.

MAJOR COMMENTS

• Not distinguishing some indication of high-risk opioid prescribing (e.g., long-term or high dose) from any prescription opioid receipt seems like an important limitation, given that the introduction frames potential adverse outcomes (nonmedical use, OUD) in terms of prevalence among patients receiving long-term opioid therapy. The authors note that dosage and duration information is not readily available prior to April 2014 – might it be possible to analyze a subset of the data for which these measures are available to gain some insight into patterns of high-risk prescriptions (perhaps just looking at characteristics associated with high-risk prescriptions, if there is not enough longitudinal data to examine trajectories over time)?

• Related to the above point, the introduction could make a better case for why it is useful to examine trajectories of any prescription opioid receipt in this population (e.g., are there data demonstrating increased risk for adverse outcomes resulting from any opioid exposure in this population)? The section of the introduction stating “one population particularly vulnerable to deleterious effects of opioid use is individuals with a history of interpersonal violence” could be expanded in this regard. Additionally, the need for research examining high-risk opioid prescriptions in this population deserves more attention in the limitations section.

• Please explain in the methods why baseline was defined differently for patients who had ≥1 opioid prescription (date of first prescription) and patients who had none (date of first EHR entry), rather than defining consistently as the date of first EHR entry during the study period for all patients.

• It is unclear to me in the methods whether referral for interpersonal violence services preceded the measured trajectory of opioid receipt. Did the referral always precede the baseline date (first opioid prescription or EHR entry during the study period), or could patients have entered the cohort prior to being referred?

• The variables section of the methods should more clearly state that the outcome was defined as binary indicator of any prescription opioid receipt in each month.

• Please provide more information about how smoking and illicit drug use history were measured – are these routinely screened for and documented in this health care setting?

• The implications highlighted in the discussion do not clearly follow from these findings. For example, the authors argue that history of interpersonal violence should be considered when assessing for opioid risk and in pain management care – this is presumably because people with a history of interpersonal violence are at higher risk for adverse opioid-related outcomes than those without such a history, however that was not examined in this study. What are the implications of findings from the present study—which examined trajectories of any prescription opioid receipt and associated baseline characteristics among people with history of interpersonal violence—for clinical care?

MINOR COMMENTS:

• Abstract, line 15: I suggest clarifying here that classes are characterized by “probability of any prescription opioid receipt at the start and end of the study period” and “change in probability over time”

• Introduction, line 23: I think this is meant to say “opioid use disorder” instead of “opioid disorder”

• Introduction, lines 41-43: Please clarify whether interpersonal violence is associated with increased likelihood of these outcomes.

Reviewer #2: The authors estimated trajectories of prescription opioid receipt over a period of 5 years among a large cohort of patients with histories of interpersonal violence. Using latent class growth analysis, they found 6 patterns of prescription opioid receipt; patterns with higher probability of opioid receipt had higher proportions of men, chronic pain diagnoses, mental health and SUD diagnoses. The authors conclude that their findings highlight the importance of accounting for these factors in opioid risk assessments and pain management plans. This exploratory analysis adds to the limited literature on prescription opioid use among people with IPV, and the paper is well-written. The conclusions and clinical importance of this work is less clear, especially because no dosage information was available and the design did not include any outcome assessment. I believe the paper could be strengthened by more clearly connecting the findings to specific clinical implications.

Major comments

Well-written paper addressing gaps in knowledge about prescription opioid use among people with histories of IPV.

Were models looking at frequency or number of opioid prescriptions, or at something else? At what time intervals? It is not clear to me exactly what “high, medium, low, etc.” means here, which limits the ability to interpret the clinical relevance of the findings.

I believe the paper would be stronger if the authors could more directly explain how their findings add new understanding that will inform clinical practice. It is notable that such a high proportion of this cohort with histories of IPV had received an opioid prescription, that mental health and substance -related diagnoses were so prevalent, and that these comorbidities were (not surprisingly) more prevalent among classes with higher probability of receiving opioid prescriptions. What do we take away from this? I appreciate the discussion about synergism and the need for better screening tools/practices. Perhaps one implication is that when treating people with chronic pain and IPV, it may be particularly important to monitor closely and to offer treatment that is trauma-informed, destigmatizing and integrated, so that synergistic comorbidities are addressed.

Minor comments

It seems all data was electronically abstracted from the EHR, but I do not see that this is made explicit. Please clarify.

Would be helpful to characterize the cohort by type of IPV–in particular, how many were older /vulnerable adults vs people who had experienced intimate partner violence/sexual trauma.

In Table 1, under the class analysis, the percentages are based on the total cohort of 1587. I wonder if these should be calculated based on the number of participants who received an opioid rx and were included in the class analysis.

Line 160: the text states that 70.8% of participants were in the never + class 6 groups; however, the percentages in those groups do not sum to 70%. This may need correction or clarification.

The lack of dose/MED is a significant limitation. While noted in the limitations section, I wonder if the authors could discuss this further. What specifically is the clinical relevance of the classes that were identified by their model, and how does the lack of MED data affect the clinical implications?

Does the calculation of standardized differences allow for calculations of significance and for adjustment for confounders? I am not sure what to make of differences in prevalence alone. Especially regarding gender differences, as the numbers were small.

Line 217: prescription opioid users could be rephrased to more patient-centered language (“people who use prescription opioids”)

How do the authors interpret the prevalence of opioid related diagnoses in the different classes, given that people with OUD at baseline were excluded: does this suggest that these patients developed OUD or simply reflect better capture of diagnoses, or both?

Line 258: Given that the data does not include any development of OUD or other outcomes, I’m not sure the findings justify this statement.

Line 259: Please change "abuse" to "use"

I believe the conclusions (final paragraph) are overstated and recommend that they be reworded. Without dose information, I’m not sure this can be considered “comprehensive.” And I’m not convinced that the authors have described how “understanding profiles of risk informs clinical screening protocols and precision of clinical treatment plans…” I think the key questions are how these risk profiles translate into improving appropriate / safety of opioid prescribing and chronic pain management in general for this population.

6. PLOS authors have the option to publish the peer review history of their article (what does this mean?). If published, this will include your full peer review and any attached files.

Reviewer #1: No

Reviewer #2: No

---

## [Author Response · Author response to Decision Letter 0]

20 Jul 2022

See attached Response to Reviewers Letter

---

## [Editor Report · Decision Letter 1]

17 Aug 2022

Trajectories and correlates of opioid prescription receipt among patients experiencing interpersonal violence

PONE-D-22-04197R1

Dear Dr. Williams,

We’re pleased to inform you that your manuscript has been judged scientifically suitable for publication and will be formally accepted for publication once it meets all outstanding technical requirements.

Kind regards,

Judith I Tsui

Academic Editor

PLOS ONE
---

## [Editor Report · Acceptance letter]

30 Aug 2022

PONE-D-22-04197R1 

Trajectories and correlates of opioid prescription receipt among patients experiencing interpersonal violence 

Dear Dr. Williams:

I'm pleased to inform you that your manuscript has been deemed suitable for publication in PLOS ONE. Congratulations! Your manuscript is now with our production department. 

Kind regards, 

on behalf of

Dr. Judith I Tsui 

Academic Editor

PLOS ONE